# Is this Written By AI?

## Abstract

With the rapid advancement of large language models (LLMs) and generative AI technology, a challenging issue has emerged: How can we determine whether an article on the internet was written by a real person or generated by an LLMs-based AI? As the barriers to training and inference of LLMs continue to lower, a vast number of AI-generated articles could enable an inexperienced person to cheat as an expert in a particular field. Traditional text plagiarism detection techniques can address this issue to some extent, but all of them have their own limitations. In this paper, we provide a systematic review of existing text plagiarism detection methods and propose a new benchmark to evaluate the accuracy of various text plagiarism detection techniques across different scenarios.

## 1 Challenges

Large Language Models (LLMs) Vaswani et al. (2017); Jaderberg et al. (2015); Radford (2018); Devlin (2018); Brown et al. (2020) can be considered one of the most significant technological breakthroughs in the field of AI in recent years. Taking conversational agents like ChatGPT as an example, people have suddenly realized that AI can now make fluent conversations with humans in different scenarios. You can interact with ChatGPT using natural language, allowing it to act as your personal assistant to help you solve various problems, such as researching information, making plans, writing articles, and more. In 2014, a Hollywood movie named Interstellar was released, featuring a "box-shaped" conversational robot - Tars, which left a deep impression on many viewers. At that time, such a conversational robot existed only in the science fiction film. However, with the advancement of LLMs technology, Tars has now become a reality in people's lives.

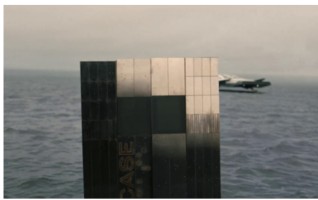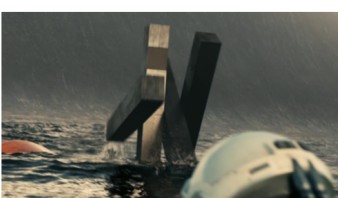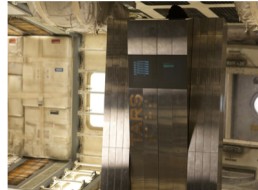

Figure 1: Tars conversational robot in Interstellar.

The success of LLMs relies on breakthroughs in three key areas: First, a series of deep neural network models Krizhevsky et al. (2012); He et al. (2016); Sutskever (2014), represented by the Transformer Vaswani et al. (2017), have rapidly advanced in recent years. Second, computing power providers, exemplified by Nvidia, have increased the computational capacity for deep learning by several orders of magnitude. Finally, over the past decade of rapid development in mobile internet, the entire internet has accumulated an enormous amount of human-generated data available for AI training. The convergence of these three factors has led to a qualitative leap in AI technology. Specifically, starting from the Turing test proposed by Alan Turing, the field of natural language processing — a decades-long unsolved challenge in computer science - has achieved groundbreaking progress.

However, with the rapid advancement of LLMs technology, a challenging issue has emerged: If AI truly passes the Turing test, how should we determine whether an account posting online is backed

by a real person or a LLMs-based AI conversational agent? How should we discern whether an article on the internet was written by a human or generated by an LLMs-based AI?

Consider the following scenario: A person who has never systematically studied medical knowledge and nor has any clinical experience uses AI tools like ChatGPT to generate a large number of healthcare-related articles, then presents themselves as a medical expert on the internet. In matters closely related to life and health, how can we promptly identify that these articles were generated by AI rather than written by the so-called "medical expert"?

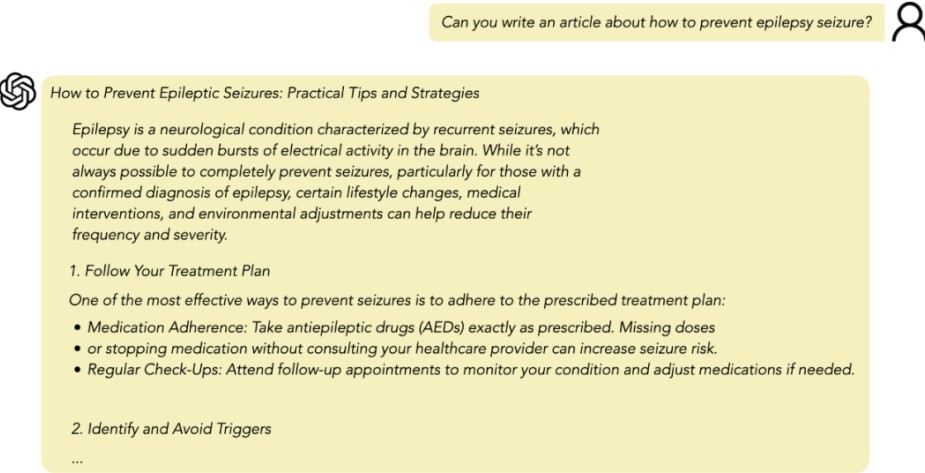

Figure 2: Requiring ChatGPT to write an article about how to prevent epilepsy seizures.

The most straightforward way for an account to prove that it belongs to a real user rather than an AI user is through real-name authentication. For instance, during the account registration process, users are required to submit various forms of identification information that are linked to their real-world identity, such as their ID card and mobile phone number. In addition, some apps also require users to provide biometric data for authentication, such as facial recognition and fingerprint information. With this real-name data, we can at least ensure that a real-name account is matched to an actual individual. This way, if any suspicious activity is detected on a real-name account, regulatory authorities can investigate and take action against the user in the real world.

However, relying solely on real-name authentication is insufficient to address the issues we previously mentioned. For example, consider the case where someone uses ChatGPT to generate a large number of healthcare-related articles, presenting themselves as a medical expert. When it comes to matters of life and health, simply relying on internet-based real-name authentication and addressing problems after they arise may already be too late. How can we detect that these articles are AI-generated at the earliest possible stage? Current solutions to this problem primarily focus on the research direction of text plagiarism detection within the field of information retrieval. In the next few sections, we will give a systematic review of existing text plagiarism detection methods.

## 2 MINIMUM EDIT DISTANCE

We first define what is text plagiarism: Suppose text $D1$ consists of $S1$ characters, and text $D2$ consists of $S2$ characters. The minimum edit distance between $D1$ and $D2$ is $Q$, which refers to the minimum number of edit operations required to transform text $D1$ to $D2$. Edit operations include adding a character, deleting a character, and modifying a character. We seek to find an $\eta$ that satisfies:

$$Q - \eta = |S_2 - S_1|, \eta \geq 0 \tag{1}$$

Then, the probability $p$ of plagiarism between the two articles is:

$$p = 1 - \frac{\eta}{S_1} \tag{2}$$

In other words, $p$ represents the degree of suspicion of plagiarism between $D1$ and $D2$. The closer $p$ is to 1, the higher the likelihood of plagiarism between $D1$ and $D2$. Conversely, the closer $p$ is to 0, the lower the likelihood of plagiarism between $D1$ and $D2$. We use an excerpt from the paper *"Attention Is All You Need"* as an example, as shown in Figure 3:

Figure 3: $D1$ is the original article, while D2 to D8 are articles modified through different methods. We consider that all of the aforementioned seven methods constitute plagiarism.

In $D2$, the plagiarist directly copies the content of $D1$ without any modifications, meaning the minimum edit distance between $D1$ and $D2$ is $Q = 0$, and $|S2 - S1| = 0$. By setting $\eta$ to 0, the condition $Q - \eta = |S2 - S1|$ is satisfied. Through calculation, the plagiarism suspicion degree $p = 1$, indicating that the two articles constitute plagiarism.

In $D3$, the plagiarist modifies only one word in $D1$ (change *that* to *which*). The calculated minimum edit distance $Q$ between $D1$ and $D3$ is 4, and $|S2 - S1| = 1$. By setting $\eta$ to 3, the condition $Q - \eta = |S2 - S1|$ is satisfied. At this point, the plagiarism suspicion degree $p = 0.9888$, suggesting a very high likelihood of plagiarism between $D1$ and $D3$.

In $D4$, the plagiarist not only modifies one word in $D1$ but also adds a new section of text. The calculated minimum edit distance $Q$ between $D1$ and $D4$ is 332, and $|S2 - S1| = 329$. By setting $\eta$ to 3, the condition $Q - \eta = |S2 - S1|$ is satisfied. The plagiarism suspicion degree $p = 0.9888$, again indicating a very high likelihood of plagiarism between $D1$ and $D4$, even though the plagiarist added new content to $D4$.

In $D5$, the plagiarist deletes a paragraph from $D1$. The calculated minimum edit distance $Q$ between $D1$ and $D5$ is 98, and $|S2 - S1| = 98$. By setting $\eta$ to 0, the condition $Q - \eta = |S2 - S1|$ is satisfied. The plagiarism suspicion degree $p = 1$, meaning the two articles constitute plagiarism.

In $D6$, the plagiarist deletes a paragraph from $D1$ and modifies one word in the remaining text. The calculated minimum edit distance $Q$ between $D1$ and $D6$ is 101, and $|S2 - S1| = 97$. By setting $\eta$ to 4, the condition $Q - \eta = |S2 - S1|$ is satisfied. The plagiarism suspicion degree $p = 0.9851$, again suggesting a very high likelihood of plagiarism between the two articles.

In $D7$, the plagiarist makes the same modifications as in $D6$ and adds a new section of content. The calculated minimum edit distance $Q$ between $D1$ and $D7$ is 268, and $|S2 - S1| = 231$. By setting $\eta$ to 37, the condition $Q - \eta = |S2 - S1|$ is satisfied. The plagiarism suspicion degree $p = 0.8624$.

Whether the two articles constitute plagiarism at this point depends on the strictness of our detection criteria. Typically, we consider $p > 0.8$ as indicative of plagiarism.

In $D8$, the plagiarist makes a deep modifications in original text. The calculated minimum edit distance $Q$ between $D1$ and $D8$ is 114, and $|S2 - S1| = 11$. By setting $\eta$ to 103, the condition $Q - \eta = |S2 - S1|$ is satisfied. At this time, the plagiarism suspicion degree $p = 0.5726$. This indicates that the detection algorithm does not consider $D8$ and $D1$ to constitute plagiarism. The detection algorithm fails at this time if we use $p > 0.8$ as our baseline.

The calculation of the minimum edit distance is a classic dynamic programming problem, with a time complexity of $O(S_1 \times S_2)$ and a space complexity of $O(S_1 \times S_2)$. In the context of LLMs, the primary issue with using the minimum edit distance algorithm is that we need to store every generated article in a database for future plagiarism checks. This would impose a significant storage overhead on services like ChatGPT. Therefore, alternative solutions for text plagiarism detection are necessary.

## 3 HASH ENCODING

Due to the storage overhead issues associated with the minimum edit distance method, another approach for text plagiarism detection is to use hash encoding. Specifically, when an LLMs service like ChatGPT generates an article, it performs a hash operation on the generated text string to obtain its hash code, which is then stored on ChatGPT's database. Subsequently, whenever a user publishes a new article on a platform, the platform's backend service performs a hash operation on the newly published article and requests ChatGPT (or other mainstream LLMs service providers) to check whether the same hash code exists in its database. If a match is found, it indicates a very high likelihood that the article is plagiarized.

Hash-based text plagiarism detection offers several advantages, the most direct being its minimal storage requirement. For example, using the SHA-256 Gilbert & Handschuh (2003) hash algorithm, regardless of the size of the input text, the resulting hash code is only 256 bits. If the MD5 Rivest (1992) hash algorithm is used, the resulting hash code is only 128 bits. Compared to storing the full text of every article for minimum edit distance calculations, this approach significantly reduces storage overhead for services like ChatGPT.

To prevent hash collisions due to the vast volume of generated text on the internet, we can employ a chunk-based hash approach. This involves dividing a text into multiple segments, performing a hash operation on each segment individually, and then combining the hashes of all segments to form the hash of the entire text, as shown in Figure 4.

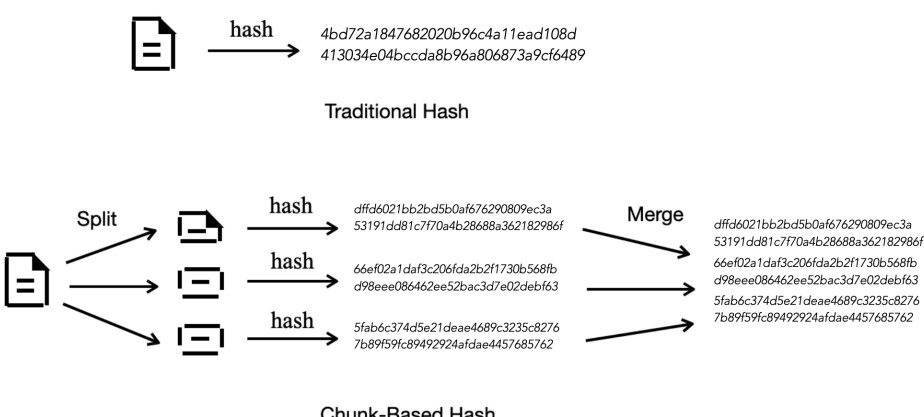

Figure 4: Traditional hash and chunk-based hash.

Despite the advantage of reduced storage space, hash algorithms have an inherent characteristic: even the slightest change in the input text (even a single byte) results in a completely different hash

code. For instance, a plagiarist can bypass hash-based detection by simply modifying a single word or punctuation mark in the text. Returning to the example of the paper *"Attention Is All You Need"*, as shown in Figure 3 with the case of D3, a plagiarist can easily evade hash-based detection by altering just one word in the article. Due to this limitation, hash-based text similarity detection is typically suited for identifying simple, programmatic, large-scale plagiarism. Examples include using tools like web crawler to scrape and republish existing text or programmatically generating articles in bulk via ChatGPT's API. Using Hash-based detection methods typically necessitates some preliminary preprocessing of the text, such as tokenizing the input article and removing punctuations.

## 4 DOCUMENT SIMILARITY

To address the limitation of hash-based detection methods, which are "unable to detect minor modifications", document similarity detection is another common approach for plagiarism detection. This method can, to some extent, solve the issue where plagiarists attempt to bypass detection by making minor changes to a few words in the text. The specific approach is as follows: First, an article is vectorized. For example, each word appearing in the article is treated as a feature in the vector space, and the frequency of each word is used as the value of the corresponding feature. Then, the feature vectors of the two articles to be compared are analyzed for similarity. For instance, cosine similarity Xia et al. (2015) can be used for comparison. This method evaluates the similarity between two word vectors by measuring the cosine of the angle between them, mathematically represented as:

$$\cos \theta = \frac{A \cdot B}{|A| \times |B|} \tag{3}$$

If two articles are identical, meaning vectors A and B are exactly the same, then $Similarity = 1$. The closer the similarity is to 1, the more similar the two articles are. If the two articles share no common words, meaning vectors $A$ and $B$ are orthogonal, then $Similarity = 0$. The closer the similarity is to 0, the less similar the two articles are. Using the texts from Figure 3 as an example, we calculate the cosine similarity for the text $D2$ to $D8$ with $D1$, and the results are as follows: $0.9999, 0.9799, 0.7780, 0.8538, 0.8221, 0.5982, 0.8011$.

In addition to using vector space comparison methods like cosine similarity, Jaccard Similarity Bag et al. (2019) is another commonly used method for similarity comparison. This method places the words from two articles into two sets (with duplicate words removed) and calculates the ratio of the intersection to the union of the two sets to determine their similarity. It is mathematically represented as:

$$J(A, B) = \frac{|A \cap B|}{|A \cup B|} \tag{4}$$

If two articles are identical, meaning set $A$ and set $B$ are exactly the same, then $Similarity = 1$. The closer the similarity is to 1, the more similar the two articles are. Conversely, if the two articles share no common words, meaning the intersection between set $A$ and set $B$ is empty, then $Similarity = 0$. The closer the similarity is to 0, the less similar the two articles are. Using the texts from Figure 3 as an example, we calculate the jaccard similarity for $D2$ to $D8$ with $D1$, and the results are as follows: $1.0, 0.9285, 0.4062, 0.7407, 0.6785, 0.3125, 0.5$.

Compared to hash-based detection, similarity-based methods can, to some extent, address the issue of plagiarists bypassing detection through minor modifications. However, the drawback is that they require more storage space to store the word vectors of each article.

Compared to minimum edit distance method, similarity-based methods still saves significant storage overhead compared to storing the entire article. However, similarity-based detection methods exhibit lower judgment accuracy in certain scenarios compared to the minimum edit distance algorithm. For instance, Cosine similarity failed to reach the 0.8 baseline in both scenarios $D4$ and $D7$. Jaccard similarity failed to reach the 0.8 baseline in the five scenarios of $D4$, $D5$, $D6$, $D7$, and $D8$. In contrast, the minimum edit distance method only failed to reach the 0.8 baseline in scenario $D8$. Based on these results, we conjecture that cosine similarity outperforms the other two detection methods in deep modifications. Meanwhile, the minimum edit distance method outperforms the

other two detection methods in scenarios involving deletions and additions to the texts. To validate our conjecture, we built a benchmark and conducted experiments on the aforementioned different detection methods in the following section.

# 5 EXPERIMENTS

In this section, we will construct a benchmark to evaluate the accuracy of various text plagiarism detection methods across the 8 scenarios depicted in Figure 3. We have initially selected 10 high-impact papers from the fields of machine learning and deep learning, with the list of papers as follows:

| | | |
|---|---|---|
| paper-1: | Deep residual learning for image recognition. | *Cited by: 255149* |
| paper-2: | Adam: A Method for Stochastic Optimization. | *Cited by: 202623* |
| paper-3: | ImageNet Classification with Deep Convolutional Neural Networks. | *Cited by:138265* |
| paper-4: | Random Forest. | *Cited by: 147177* |
| paper-5: | Very Deep Convolutional Networks for Large-Scale Image Recognition. | *Cited by: 137497* |
| paper-6: | Scikit-learn: Machine Learning in Python. | *Cited by: 105781* |
| paper-7: | Attention Is All You Need. | *Cited by: 151961* |
| paper-8: | Support-Vector Networks. | *Cited by: 72612* |
| paper-9: | Generative Adversarial Nets. | *Cited by: 77545* |
| paper-10: | Faster-RCNN: Towards Real-Time Object Detection With Region Proposal Networks. | *Cited by: 36077* |

Figure 5: Selected 10 high-impact papers in machine learning and deep learning. The citations is updated in 2025-2-9.

Here, we take the abstract of the paper "Deep Residual Learning for Image Recognition" as an example to illustrate how we construct our testing benchmark, as shown in Figure 6.

It is important to note the distinction between $D3$ and $D8$, both of which are modifications of $D1$. However, $D3$ involves only simple word substitutions based on $D1$, changes that could be made by someone entirely unfamiliar with the content of the article. In contrast, $D8$ entails a comprehensive revision of $D1$, requiring the modifier to possess a certain level of understanding of the original article's content as well as background knowledge in the relevant field.

We modified the abstracts of the selected 10 papers according to the seven methods from $D2$ to $D8$, and then performed text plagiarism detection on them using the minimum edit distance, cosine similarity, and jaccard similarity, respectively. Here, we artificially consider all modifications as constituting plagiarism, meaning that the detection algorithm's judgment probability must exceed 0.8 to be considered a successful detection. The final experimental results are shown in Figure 7.

Each detection method was subjected to 70 tests. The minimum edit distance method successfully passed 61 out of the 70 tests, achieving a detection success rate of 87.1%. Cosine similarity successfully passed 59 out of the 70 tests, with a detection success rate of 84.2%. Jaccard similarity, however, only passed 21 out of the 70 tests, resulting in a test success rate of merely 30%.

Further analysis reveals that all detection methods passed the case of D3, meaning that if someone with no understanding of the article's content merely modifies it through simple word substitutions, it will be detected and considered as plagiarism.

We also found that the minimum edit distance outperforms Cosine Similarity and Jaccard Similarity because the way it calculates the probability $p$ is independent of the added or deleted text. If an article has numerous additions or deletions, it can cause significant discrepancies in the word vectors for cosine similarity and jaccard similarity compared to the original, leading the detection algorithms to conclude that the two articles are not plagiarized, when in fact, the plagiarized content still exists within the new text. In the calculation of the minimum edit distance, as shown in formulas (1) and (2), even if the new text includes a lot of additional content, which increases the value of $S2$, this will also increase the value of the edit distance $Q$, while $\eta$ remains unchanged. Therefore, the ratio of $\eta$ to $S1$ does not change with the addition of new content. Conversely, if the new text simply deletes a portion of the original text, it similarly does not alter the ratio of $\eta$ to $S1$.

324
325
326
327
328
329
330
331
332
333
334
335
336
337
338
339
340
341
342
343
344
345
346
347
348
349
350
351
352
353
354
355
356
357
358
359
360
361
362
363
364
365

*D1:* Deeper neural networks are more difficult to train. We present a residual learning framework to ease the training of networks that are substantially deeper than those used previously. We explicitly reformulate the layers as learning residual functions with reference to the layer inputs, instead of learning unreferenced functions. We provide comprehensive empirical evidence showing that these residual networks are easier to optimize, and can gain accuracy from considerably increased depth. On the ImageNet dataset we evaluate residual nets with a depth of up to 152 layers—8× deeper than VGG nets [40] but still having lower complexity. An ensemble of these residual nets achieves 3.57% error on the ImageNet test set. This result won the 1st place on the ILSVRC 2015 classification task. We also present analysis on CIFAR-10 with 100 and 1000 layers. The depth of representations is of central importance for many visual recognition tasks. Solely due to our extremely deep representations, we obtain a 28% relative improvement on the COCO object detection dataset. Deep residual nets are foundations of our submissions to ILSVRC & COCO 2015 competitions1, where we also won the 1st places on the tasks of ImageNet detection, ImageNet localization, COCO detection, and COCO segmentation.

*D2:* Deeper neural networks are more difficult to train. We present a residual learning framework to ease the training of networks that are substantially deeper than those used previously. We explicitly reformulate the layers as learning residual functions with reference to the layer inputs, instead of learning unreferenced functions. We provide comprehensive empirical evidence showing that these residual networks are easier to optimize, and can gain accuracy from considerably increased depth. On the ImageNet dataset we evaluate residual nets with a depth of up to 152 layers—8× deeper than VGG nets [40] but still having lower complexity. An ensemble of these residual nets achieves 3.57% error on the ImageNet test set. This result won the 1st place on the ILSVRC 2015 classification task. We also present analysis on CIFAR-10 with 100 and 1000 layers. The depth of representations is of central importance for many visual recognition tasks. Solely due to our extremely deep representations, we obtain a 28% relative improvement on the COCO object detection dataset. Deep residual nets are foundations of our submissions to ILSVRC & COCO 2015 competitions1, where we also won the 1st places on the tasks of ImageNet detection, ImageNet localization, COCO detection, and COCO segmentation.

*D3:* Deeper learning are more difficult to train. We present a residual learning framework for easing the training of networks that are substantially deeper than those used previously. We explicitly reformulate the layers as learning residual functions with reference to the layer inputs, instead of learning unreferenced functions. We also provide comprehensive empirical evidence showing that these residual networks are easier to optimize, and can gain accuracy from considerably increased depth. On the ImageNet dataset we evaluate residual networks with a depth of up to 152 layers—8× deeper than VGG nets [40] but still having lower complexity. An ensemble of these residual nets achieves 3.57% error on the ImageNet test set. This result won the first place on the ILSVRC 2015 classification task. We also present analysis on CIFAR-10 with 100 and 1000 layers. The depth of representations is of central importance for many visual recognition tasks. Due to our extremely deep representations, we obtain a 28% relative improvement on the COCO object detection dataset. Deep residual networks are foundations of our submissions to ILSVRC & COCO 2015 competitions1, where we also won the first places on the tasks of ImageNet detection, ImageNet localization, COCO detection, and COCO segmentation.

*D5:* Deeper neural networks are more difficult to train. We present a residual learning framework to ease the training of networks that are substantially deeper than those used previously. We explicitly reformulate the layers as learning residual functions with reference to the layer inputs, instead of learning unreferenced functions. We provide comprehensive empirical evidence showing that these residual networks are easier to optimize, and can gain accuracy from considerably increased depth. On the ImageNet dataset we evaluate residual nets with a depth of up to 152 layers—8× deeper than VGG nets [40] but still having lower complexity. An ensemble of these residual nets achieves 3.57% error on the ImageNet test set. This result won the 1st place on the ILSVRC 2015 classification task. We also present analysis on CIFAR-10 with 100 and 1000 layers. The depth of representations is of central importance for many visual recognition tasks. ~~Solely due to our extremely deep representations, we obtain a 28% relative improvement on the COCO object detection dataset. Deep residual nets are foundations of our submissions to ILSVRC & COCO 2015 competitions1, where we also won the 1st places on the tasks of ImageNet detection, ImageNet localization, COCO detection, and COCO segmentation.~~

*D4:* Deeper learning are more difficult to train. We present a residual learning framework for easing the training of networks that are substantially deeper than those used previously. We explicitly reformulate the layers as learning residual functions with reference to the layer inputs, instead of learning unreferenced functions. We also provide comprehensive empirical evidence showing that these residual networks are easier to optimize, and can gain accuracy from considerably increased depth. On the ImageNet dataset we evaluate residual networks with a depth of up to 152 layers—8× deeper than VGG nets [40] but still having lower complexity. An ensemble of these residual nets achieves 3.57% error on the ImageNet test set. This result won the first place on the ILSVRC 2015 classification task. We also present analysis on CIFAR-10 with 100 and 1000 layers. The depth of representations is of central importance for many visual recognition tasks. Due to our extremely deep representations, we obtain a 28% relative improvement on the COCO object detection dataset. Deep residual networks are foundations of our submissions to ILSVRC & COCO 2015 competitions1, where we also won the first places on the tasks of ImageNet detection, ImageNet localization, COCO detection, and COCO segmentation. Recent work in unsupervised feature learning and deep learning has shown that being able to train large models can dramatically improve performance. In this paper, we consider the problem of training a deep network with billions of parameters using tens of thousands of CPU cores. We have developed a software framework called DistBelief that can utilize computing clusters with thousands of machines to train large models. Within this framework, we have developed two algorithms for large-scale distributed training: (i) Downpour SGD, an asynchronous stochastic gradient descent procedure supporting a large number of model replicas, and (ii) Sandblaster, a framework that supports for a variety of distributed batch optimization procedures, including a distributed implementation of L-BFGS. Downpour SGD and Sandblaster L-BFGS both increase the scale and speed of deep network training. We have successfully used our system to train a deep network 100x larger than previously reported in the literature, and achieves state-of-the-art performance on ImageNet, a visual object recognition task with 16 million images and 21k categories. We show that these same techniques dramatically accelerate the training of a more modestly sized deep network for a commercial speech recognition service. Although we focus on and report performance of these methods as applied to training large neural networks, the underlying algorithms are applicable to any gradient-based machine learning algorithm.

*D6:* Deeper learning are more difficult to train. We present a residual learning framework for easing the training of networks that are substantially deeper than those used previously. We explicitly reformulate the layers as learning residual functions with reference to the layer inputs, instead of learning unreferenced functions. We also provide comprehensive empirical evidence showing that these residual networks are easier to optimize, and can gain accuracy from considerably increased depth. On the ImageNet dataset we evaluate residual networks with a depth of up to 152 layers—8× deeper than VGG nets [40] but still having lower complexity. An ensemble of these residual nets achieves 3.57% error on the ImageNet test set. This result won the first place on the ILSVRC 2015 classification task. We also present analysis on CIFAR-10 with 100 and 1000 layers. The depth of representations is of central importance for many visual recognition tasks. ~~Due to our extremely deep representations, we obtain a 28% relative improvement on the COCO object detection dataset. Deep residual networks are foundations of our submissions to ILSVRC & COCO 2015 competitions1, where we also won the first places on the tasks of ImageNet detection, ImageNet localization, COCO detection, and COCO segmentation.~~

*D8:* Deep learning models are harder to train as their depth increases. In this work, we introduce a residual learning approach to facilitate the training of networks much deeper than those typically used. Instead of learning unreferenced functions, we explicitly define the layers to learn residual functions based on their input layers. We provide extensive experimental results demonstrating that residual networks are easier to optimize and can achieve higher accuracy with significantly increased depth. On the ImageNet dataset, we test residual networks with depths up to 152 layers—8 times deeper than VGG networks, yet with lower complexity. An ensemble of these residual networks reaches an error rate of 3.57% on the ImageNet test set. This result secured first place in the ILSVRC 2015 classification challenge. We also analyze performance on CIFAR-10 with networks having 100 and 1000 layers. The depth of representation is crucial for a variety of visual recognition tasks. Thanks to the extremely deep representations, we achieve a 28% relative improvement on the COCO object detection dataset. These deep residual networks are the foundation of our submissions to the ILSVRC & COCO 2015 competitions, where we also took first place in ImageNet detection, ImageNet localization, COCO detection, and COCO segmentation tasks.

*D7:* Deeper learning are more difficult to train. We present a residual learning framework for easing the training of networks that are substantially deeper than those used previously. We explicitly reformulate the layers as learning residual functions with reference to the layer inputs, instead of learning unreferenced functions. We also provide comprehensive empirical evidence showing that these residual networks are easier to optimize, and can gain accuracy from considerably increased depth. On the ImageNet dataset we evaluate residual networks with a depth of up to 152 layers—8× deeper than VGG nets [40] but still having lower complexity. An ensemble of these residual nets achieves 3.57% error on the ImageNet test set. This result won the first place on the ILSVRC 2015 classification task. We also present analysis on CIFAR-10 with 100 and 1000 layers. The depth of representations is of central importance for many visual recognition tasks. ~~Due to our extremely deep representations, we obtain a 28% relative improvement on the COCO object detection dataset. Deep residual networks are foundations of our submissions to ILSVRC & COCO 2015 competitions1, where we also won the first places on the tasks of ImageNet detection, ImageNet localization, COCO detection,~~ and COCO segmentation. Recent work in unsupervised feature learning and deep learning has shown that being able to train large models can dramatically improve performance. In this paper, we consider the problem of training a deep network with billions of parameters using tens of thousands of CPU cores. We have developed a software framework called DistBelief that can utilize computing clusters with thousands of machines to train large models. Within this framework, we have developed two algorithms for large-scale distributed training: (i) Downpour SGD, an asynchronous stochastic gradient descent procedure supporting a large number of model replicas, and (ii) Sandblaster, a framework that supports for a variety of distributed batch optimization procedures, including a distributed implementation of L-BFGS. Downpour SGD and Sandblaster L-BFGS both increase the scale and speed of deep network training. We have successfully used our system to train a deep network 100x larger than previously reported in the literature, and achieves state-of-the-art performance on ImageNet, a visual object recognition task with 16 million images and 21k categories. We show that these same techniques dramatically accelerate the training of a more modestly sized deep network for a commercial speech recognition service. Although we focus on and report performance of these methods as applied to training large neural networks, the underlying algorithms are applicable to any gradient-based machine learning algorithm.

366
367
368
369
370

Figure 6: $D1$ is the original abstract of the paper; $D2$ is a direct copy of $D1$; $D3$ has some minor modifications based on $D1$; $D4$ has an additional paragraph added to $D3$; $D5$ has a paragraph removed from $D1$; $D6$ has a paragraph removed from $D3$; $D7$ has an additional paragraph added to $D6$; $D8$ is a rewritten version of $D1$ while preserving the original meaning. All the 10 selected papers (only use its abstract) will be modified through these 7 methods.

371
372
373
374
375
376
377

However, we observed that in the case of $D8$, the performance of the minimum edit distance is not as effective as that of cosine similarity. That is to say, if an article is rewritten while maintaining the original meaning, it is likely to evade detection by the minimum edit distance. For instance, the sentences "Hello Lucy" and "Lucy Hello" convey essentially the same meaning, but the alteration in word order can lead to significant differences in the calculation of the minimum edit distance, while

|    | D3 | D4 | D5 | D6 | D7 | D8 |
|----|----|----|----|----|----|----|
| P1 | min: 0.9705
cos: 0.9833
Jac: 0.9504 | min: 0.9705
cos: 0.8662
Jac: 0.5178 | min: 1.0
cos: 0.9473
jac: 0.8151 | min: 0.9721
cos: 0.9311
jac: 0.7768 | min: 0.9062
cos: 0.8094
jac: 0.4375 | min: 0.6560
cos: 0.8809
jac: 0.4785 |
| P2 | min: 0.9972
cos: 0.9921
jac: 1.0 | min: 0.9972
cos: 0.8421
jac: 0.4790 | min: 1.0
cos: 0.8878
jac: 0.6990 | min: 0.9702
cos: 0.8620
jac: 0.6893 | min: 0.8827
cos: 0.7488
jac: 0.3488 | min: 0.7042
cos: 0.8524
jac: 0.5107 |
| P3 | min: 0.9800
cos: 0.9854
jac: 0.9454 | min: 0.9800
cos: 0.8564
jac: 0.4792 | min: 1.0
cos: 0.9279
jac: 0.7358 | min: 0.9800
cos: 0.8965
jac: 0.6972 | min: 0.8872
cos: 0.7935
jac: 0.3703 | min: 0.7524
cos: 0.8770
jac: 0.5507 |
| P4 | min: 0.9900
cos: 0.9961
jac: 0.9565 | min: 0.9900
cos: 0.8782
jac: 0.4292 | min: 1.0
cos: 0.9466
jac: 0.7977 | min: 0.9880
cos: 0.9446
jac: 0.7692 | min: 0.9520
cos: 0.8236
jac: 0.3627 | min: 0.6087
cos: 0.8915
jac: 0.4883 |
| P5 | min: 0.9751
cos: 0.9824
jac: 0.9405 | min: 0.9751
cos: 0.8057
jac: 0.4589 | min: 1.0
cos: 0.9264
jac: 0.7244 | min: 0.9774
cos: 0.9160
jac: 0.6831 | min: 0.9073
cos: 0.7559
jac: 0.3671 | min: 0.5378
cos: 0.8338
jac: 0.4637 |
| P6 | min: 0.9715
cos: 0.9833
jac: 0.9558 | min: 0.9314
cos: 0.6979
jac: 0.3606 | min: 1.0
cos: 0.9489
jac: 0.8358 | min: 0.9866
cos: 0.9352
jac: 0.7941 | min: 0.9197
cos: 0.6663
jac: 0.3060 | min: 0.7040
cos: 0.7297
jac: 0.4468 |
| P7 | min: 0.9163
cos: 0.9391
jac: 0.8285 | min: 0.9306
cos: 0.6408
jac: 0.3224 | min: 1.0
cos: 0.8898
jac: 0.6666 | min: 0.9359
cos: 0.8432
jac: 0.5588 | min: 0.9110
cos: 0.5899
jac: 0.2651 | min: 0.6512
cos: 0.8088
jac: 0.5057 |
| P8 | min: 0.9731
cos: 0.9716
jac: 0.9629 | min: 0.9731
cos: 0.8034
jac: 0.4051 | min: 1.0
cos: 0.9420
jac: 0.7468 | min: 0.9642
cos: 0.9046
jac: 0.7160 | min: 0.9050
cos: 0.7542
jac: 0.3230 | min: 0.6860
cos: 0.8762
jac: 0.4909 |
| P9 | min: 0.9815
cos: 0.9936
jac: 0.9897 | min: 0.9815
cos: 0.8655
jac: 0.4688 | min: 1.0
cos: 0.9222
jac: 0.6326 | min: 0.9856
cos: 0.9208
jac: 0.6428 | min: 0.8844
cos: 0.7942
jac: 0.3205 | min: 0.7126
cos: 0.8897
jac: 0.5967 |
| P10 | min: 0.9764
cos: 0.9864
jac: 0.9316 | min: 0.9764
cos: 0.7881
jac: 0.5022 | min:1.0
cos: 0.9160
jac: 0.6991 | min: 0.9710
cos: 0.9019
jac: 0.6551 | min: 0.8867
cos: 0.7231
jac: 0.3710 | min: 0.8686
cos: 0.9096
jac: 0.6861 |

Figure 7: Experiment results for different plagiarism detection method. Note that, all the result for D2 is equal to 1.0, so we remove this column.

it does not affect the computation of cosine similarity. In other words, cosine similarity, which utilizes word vectors, is more representative of the theme of the article.

The reason why Jaccard similarity achieved only a 30% success rate in our tests is that its method of comparing the intersection and union of word sets cannot represent the full information of an article. For example, if the words 'machine' and 'learning' appear frequently in an article, it is highly probable that we can infer the article is related to machine learning. However, if the words 'machine' and 'learning' appear only once, the article may not necessarily be about machine learning. The deduplication process of Jaccard similarity loses the important information about the frequency of each word's occurrence. In contrast, Cosine similarity is adept at capturing this nuance.

## 6 FUTURE EXPLORATION

The various plagiarism detection methods mentioned in this paper are all classic and well-established, each with its own strengths and weaknesses in different scenarios. How to mitigate the shortcomings of these detection methods across various contexts will be an interesting research direction. For instance, could we use an ensemble approach to combine different detection algorithms for calculating the probability of plagiarism? Could we dynamically select different detection algorithms based on the relationship between the character lengths of the new text and the original text? Moreover, could we employ topic models like LDA to mine the semantics of the text? Additionally, how can we use the co-design of algorithms and systems to effectively reduce the computational and storage overhead on LLMs servers while ensuring detection effectiveness? These will all be research directions we focus on.

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
