# OpenReview forum: "Is This Written by AI?"
_ICLR.cc/2025/Workshop/BuildingTrust — Submitted to BuildingTrust_

### Official Review · Reviewer_M6Pm · 2025-03-01
**Review to Is This Written by AI?**

**Rating:** 2
**Confidence:** 5

**Review:**

This paper analyzes whether black-box model providers, assuming they stored all generated outputs, could reliably detect generated text in the wild. The authors analyze methods minimum edit distance, hash encoding and document similarity on a set of 7 (!) manually created variations of a base text.

This paper does not present an interesting contribution to research.
- The setting that black-box model providers store all generated outputs (and the limitation that the detection only covers text generated from black-box models, not open-weight LLMs) is never explicitly mentioned
- The evaluated methods are extremely basic and seem very limited
- The dataset is tiny and there is no obvious reason why such a small dataset should be used
- The paper does not mention any related work regarding watermarking, black-box and perplexity-based detection (e.g. Binoculars) etc.

---

### Official Review · Reviewer_uTXK · 2025-03-01
**Anonymous Review**

**Rating:** 6
**Confidence:** 4

**Review:**

This paper discusses the problem and importance of the detection of AI-generated content (LLMs specifically). Towards this, the paper surveys 3 retrieval-based metrics to detect AI-generated text viz. hash encoding, cosine similarity, and Jaccard similarity. The retrieval-based methods were first introduced in [1], which the paper should cite. The papers assert that current detectors are geared toward plagiarism detection but fail to mention other detectors like [2], [3], and [4]. The methods which doesn't assume a prior database of generated content are more realistic in nature. This paper discusses and benchmark on small toy dataset the retrievel-based methods.

I think the contributions are already known and bit to basic in nature. Nonetheless, the paper fits the workshop so I will give marginal accept.



---
[1] Kalpesh Krishna, Yixiao Song, Marzena Karpinska, John Wieting, & Mohit Iyyer. (2023). Paraphrasing evades detectors of AI-generated text, but retrieval is an effective defense.

[2] Eric Mitchell, Yoonho Lee, Alexander Khazatsky, Christopher D. Manning, & Chelsea Finn. (2023). DetectGPT: Zero-Shot Machine-Generated Text Detection using Probability Curvature.

[3] Abhimanyu Hans, Avi Schwarzschild, Valeriia Cherepanova, Hamid Kazemi, Aniruddha Saha, Micah Goldblum, Jonas Geiping, & Tom Goldstein. (2024). Spotting LLMs With Binoculars: Zero-Shot Detection of Machine-Generated Text.

[4] Guangsheng Bao, Yanbin Zhao, Zhiyang Teng, Linyi Yang, & Yue Zhang. (2024). Fast-DetectGPT: Efficient Zero-Shot Detection of Machine-Generated Text via Conditional Probability Curvature.

---

### Decision · Program_Chairs · 2025-03-04

Reject